# Experimental Airway Allogenic Transplantation Model with Decellularized Cryopreserved Tracheas

**DOI:** 10.3390/biomedicines13102401

**Published:** 2025-09-30

**Authors:** Néstor J. Martínez-Hernández, Lara Milián-Medina, Jorge Mas-Estellés, Amparo Roig-Bataller, David Hervás-Marín, Manuel Mata-Roig

**Affiliations:** 1Department of Thoracic Surgery, Hospital Biomédico Ascires, 46014 Valencia, Spain; 2Faculty of Biomedical and Health Sciences, Universidad Europea de Valencia, 46010 Valencia, Spain; 3Department of Thoracic Surgery, Hospital Universitari de la Ribera, 46600 Alzira, Spain; roig_ampbat@gva.es; 4Pathology Department, Medicine and Odontology Faculty, University of València, 46010 Valencia, Spain; lara.milian@uv.es (L.M.-M.); manuel.mata@uv.es (M.M.-R.); 5Center for Biomaterials and Tissue Engineering, Universitat Politècnica de València, 46022 Valencia, Spain; jmas@fis.upv.es; 6Department of Applied Statistics and Operations Research and Quality, Universitat Politècnica de València, 46022 Valencia, Spain; daherma@eio.upv.es; 7Networking Research Centre on Respiratory Diseases (CIBERER), ISCIII, 28222 Madrid, Spain

**Keywords:** trachea, airway, bioengineering, transplantation, tissue engineering, biomechanics

## Abstract

**Background/Objectives**: Tracheal replacement remains an unmet necessity in airway pathology treatment. We describe a decellularization and cryopreservation tracheal replacement protocol tested in vivo. **Methods**: A prospective experimental cohort study is conducted in which tracheas from white New Zealand rabbits (n = 32) are decellularized and cryopreserved, sterilized with gamma radiation, and tutorized with a stent. Bilateral, pedicled fasciomuscular flaps are harvested, implanting one cryopreserved and one non-cryopreserved in each of 16 rabbits, dividing them into four groups (remaining tracheas implanted at 2, 4, 8, and 12 weeks, respectively). After removal, the tracheas are both histologically and biomechanically evaluated. **Results**: Decellularization is effective, with minimal effects on the biomechanical characteristics of the trachea. Implantation results in a well-vascularized organ, with no inflammatory or tissular rejection cellular response. Organs achieve all basal histological and biomechanical characteristics within 8 weeks of implantation, with no differences observed between cryopreserved and non-cryopreserved scaffolds. **Conclusions**: The present decellularization and cryopreservation protocol yields biocompatible, well-tolerated by the host tracheas with analogous histological and biomechanical characteristics to native ones.

## 1. Introduction

Extensive tracheal resection remains one of the most complex challenges in thoracic surgery. Although the trachea may appear structurally simple, its biomechanical and functional characteristics—rigidity, elasticity, luminal patency, and mucociliary integrity—are difficult to replicate. When significant segments are affected by disease, resection becomes limited not by surgical technique but by the absence of a suitable replacement [1].

The main limitation we face when treating patients with tracheal disorders is the inability to resect as much of the organ as necessary. Several strategies have been proposed to extend the length of trachea that can be safely resected, including laryngeal and hilar release, tracheobronchial sleeve, the application of plastic techniques, or even the creation of a new trachea using autologous flaps [1,2]. Despite these alternatives, achieving a suitable tracheal replacement remains elusive.

As early as 1950, Belsey outlined the essential requirements for an effective tracheal replacement. Remarkably, these criteria remain valid today: lateral rigidity, longitudinal elasticity and flexibility, adequate luminal caliber, and uninterrupted epithelial lining [3]. Additional desirable features include immediate availability of the replacement (i.e., an ‘off-the-shelf’ solution) [4] and no need for immunosuppression, given that cancer—where immunosuppressive therapy is contraindicated—is among the main indications for extensive tracheal resection [5].

In recent years, both experimental and clinical success have been achieved in humans through the implantation of a tracheal allograft with temporary immunosuppression [6], as well as by implanting flap-wrapped aortic grafts—either cryopreserved or fresh—with endoluminal stents [7]. Nevertheless, these approaches have consistently failed to meet the criteria for an ideal replacement.

Specifically, most do not achieve full circumferential regeneration, do not lead to a self-renewing structure, often require prolonged or permanent stenting, lack immediate availability, and still necessitate immunosuppressive therapy [5,6].

A common feature among many of these approaches—due to the critical role of vascularization—is the use of heterotopic implantation, or prelamination. This phase enables the graft to develop a vascular supply prior to definitive (eutopic) implantation, with its own vascular supply [8].

In this context, we conducted the present study to assess the tolerability and feasibility of a decellularization and cryopreservation protocol in an experimental rabbit model. The aim of this protocol was to obtain a tracheal substitute with characteristics similar to those of the native organ, but without the need for immunosuppression (owing to a fully decellularized scaffold), and with the advantage of immediate availability through cryopreservation. The protocol was adapted from a previously validated porcine model [9] and was designed to assess the structural, cellular, and biomechanical integrity of the decellularized trachea both in vitro and in vivo. Secondary objectives included assessing the feasibility of cryopreservation, determining in vivo graft viability across short-, medium-, and long-term time points, and identifying the optimal prelamination duration prior to eutopic transplantation.

## 2. Materials and Methods

### 2.1. Ethics Statement

The European directive 2010/63/EU for the care and use of laboratory animals was strictly followed. The study protocol was approved by the Ethics Committee of the University of Valencia (Law 86/609/EEC and 214/1997 and Code 2018/VSC/PEA/0122 Type 2 of the government of Valencia, Spain).

### 2.2. Controls

Native tracheas were obtained from 10 adult New Zealand albino rabbits (Oryctolagus cuniculus) included in the control arm of ongoing studies at our center. Different segments of these tracheas were histologically, ultrastructurally, and biomechanically characterized (Appendix A).

### 2.3. Tracheal Decellularization

The whole tracheal decellularization process has been previously communicated [9]. Tracheas were harvested from adult male New Zealand white rabbits (3.5–4.2 kg). Each trachea was divided into 2-cm segments, yielding a total of 32 specimens. Connective tissue and inner mucosal layers were meticulously removed [5].

The decellularization process was based on a previously described protocol [10]. Briefly, specimens were immersed in phosphate-buffered saline (PBS) containing 2% sodium dodecyl sulfate (SDS; Sigma-Aldrich, St. Louis, MO, USA), 5% penicillin–streptomycin, and 5% amphotericin B (Gibco, Thermo Fisher Scientific, Waltham, MA, USA). Samples were maintained at room temperature under constant agitation for five weeks. The medium was changed weekly following a two-hour osmotic shock.

Half of the decellularized specimens underwent cryopreservation: tracheas were submerged in a cryoprotective solution consisting of 80% fetal bovine serum (GE Healthcare HyClone, Madrid, Spain) and 20% dimethyl sulfoxide (Sigma-Aldrich; MO, USA), placed in a Mr. Frosty™ freezing container (Thermo Fisher; Madrid, Spain), and stored at −80 °C for 13–15 days. Thawing was performed in a 37 °C water bath, followed by PBS rinsing. All samples were sterilized by 1 kGy gamma irradiation using a TrueBeam^®^ linear accelerator (Varian, Palo Alto, CA, USA).

### 2.4. Calculation of Sample Size (n)

Sample size (n) was calculated based on Young’s modulus, assuming a mean of 0.6 MPa, a standard deviation of 0.35 MPa, and a desired precision of ±0.1 MPa. Anticipating a 20% loss rate [11], a total of 58 samples plus 4 extra ones for potential losses were needed. These were distributed as follows: 10 control samples, 16 samples for in vitro study of decellularized tracheas (4 non-cryopreserved, 4 cryopreserved, 4 non-cryopreserved and irradiated, and 4 cryopreserved and irradiated), and 32 tracheal specimens implanted in 16 rabbits (2 samples per rabbit).

### 2.5. Implantation Technique

Surgical implantation was performed as previously described [9]. Each decellularized trachea was fitted with a sterile 14 Fr intraluminal polyvinyl chloride stent (Argyle, Medtronic; Istanbul, Turkey). Implants were placed in 16 male New Zealand white rabbits (3.6–4.1 kg). Through a 3-cm mid-thoracic longitudinal incision, bilateral pectoral flaps (fascia and muscle) were created. In each animal, one cryopreserved trachea was implanted into the right hemithorax and one non-cryopreserved trachea into the left.

Postoperative care included administration of enrofloxacin 2.5%, 0.5 mL/kg, and meloxicam 5 mg/mL, 0.05 mL/kg (both provided by Boehringer Ingelheim, Ingelheim am Rhein, Germany) every 24 h for five days. Animals were divided into four groups (n = 4 per group) for analysis at the short- (2 and 4 weeks), medium- (8 weeks), and long-term (12 weeks), as defined by ISO [10].

### 2.6. Structural Study

Standard staining was performed using hematoxylin–eosin (H&E), Masson’s trichrome (MT), and orcein (O). Decellularization was assessed via DAPI staining and DNA quantification as described previously [9]. Inflammatory infiltrates were scored in accordance with ISO norms for biological evaluation of medical devices; scores ranged from 0 to 4 depending on the findings observed per high-power field (400×) [10]. Variables assessed included eosinophils, neutrophils, plasma cells, macrophages, lymphocytes, giant cells, connective tissue formation, adipose tissue formation, and cartilage necrosis.

Vascularization was evaluated by immunohistochemistry (CD31 expression) using a specific mouse monoclonal antibody for rabbits (ab212712; Abcam; Cambridge, UK). Immunohistochemistry was performed according to standard procedures [11]. Vessel density was measured by counting the number of vessels in five different 400× fields.

For observation by scanning electron microscope (SEM), we followed the previously reported procedure [11] using a JSM-5410 SEM (Jeol USA Inc., Peabody, MA, USA). Photographs were taken at a distance of 7 mm, with a voltage of 2 kV.

### 2.7. Biomechanical Study

Two types of biomechanical tests were conducted: axial tensile and radial compression, following the method described elsewhere in all suitable samples [12].

#### 2.7.1. Tensile Tests

Tensile tests were performed with the Adamel Lhomargy DY34 universal testing machine (UTM) (Testing Machines; Veenendaal, The Netherlands) with displacement control (TestWorks 4 software; MTS Systems Corporation; Eden Prairie, MN, USA). Stretching was at a displacement rate of 5.0 mm·min^−1^, with data recording every 0.4 s.

The tracheas were attached to jaws with preformed holes using a continuous 6-0 nylon monofilament (Monosoft; Covidien; Mansfield, MA, USA). Pre-test deformation due to jaws and sutures was measured and subtracted from the final values. Variables were as follows: energy stored per unit of trachea volume (W/Vol, mJ·mm^−3^); and Young’s modulus (E, MPa). Maximal stress (σ_max_, MPa) and strain (ε_max_, unitless) were obtained from the tracheal breakpoint.

#### 2.7.2. Radial Compression Test

These tests were performed on a Microtest UTM displacement control with SCM300095 software (Microtest, Madrid, Spain). Data were recorded at 0.5 s intervals. After the tensile test, the torn region of the tracheal suture was resected, and the tracheas were placed within the membranous area resting on the lower plate, which was gradually raised towards the top plate at a constant speed of 5 mm·min^−1^.

Output variables included the force tolerated per unit of length (ƒ in N·mm^−1^), tracheal stiffness (ℛ, Mpa·mm) to radial compression, and energy per unit of surface area (W/S, mJ·mm^−2^).

### 2.8. Statistical Analysis

Ordinal regression models were used to analyze histological variables, including the interaction between cryopreservation status and implantation duration. The models were adjusted using Bayesian statistics with graphing of marginal effects for each model. Results are expressed as odds ratio (OR) with 95% credible intervals (CI).

Tensile test variables were analyzed using multiple linear regression models, including “decellularization” as a covariate and the interaction between the variables “weeks of implant” and “cryopreservation”. In the compression tests, W/S was analyzed in the same way as in the tensile tests. For ƒ and ℛ, linear mixed models were used, with the percentage of occlusion entered as a monotonic effect. These were adjusted using Bayesian statistics, with partial dependence graphics. Results are expressed as an estimation of the amount of the effect of the variables in our sample and a 95% CI. All analyses were conducted in R (v.3.5.3; https://www.R-project.org, accessed on 10 January 2025) using the clickR (v.0.4.32) and brms (v.2.9.0) packages.

## 3. Results

### 3.1. In Vitro Evaluation of Tracheal Decellularization

#### 3.1.1. Histology

DAPI staining confirmed the absence of residual cell nuclei in the decellularized tracheas, while hematoxylin–eosin (H&E) staining demonstrated complete removal of mucosal and submucosal cellular components, with only scant cellular debris remaining in the cartilage (Figure 1A). MT and O staining revealed the complete preservation of collagen and elastic fibers (Figure 1B–C). Quantification of DNA content consistently yielded values below 50 ng per mg of dry tissue, thus meeting established criteria for adequate decellularization [13]. SEM observation demonstrated no disruption of the characteristic architecture of collagen and elastic fibers (Figure 1D).

#### 3.1.2. Biomechanics

Tensile testing following decellularization revealed a non-significant trend toward reduced ε_max_ (−0.204 CI [−0.407, 0.005]) values. Significant reductions were observed in the following parameters compared to native tracheas: σ_max_ (−0.246 MPa CI [−0.348, −0.145] MPa), E (−0.408 MPa CI [−0.688, −0.13] MPa), and W/Vol (−0.124 mJ·mm^−3^ CI [−0.195, −0.055) mJ·mm^−3^].

Decellularization did not affect the compression W/S (−0.691 mJ·mm^−2^ CI [−1.419, 0.018] mJ·mm^−2^), but significantly reduced both ƒ (−0.06 N·mm^−1^ CI [−0.088, −0.035] N·mm^−1^) and ℛ (−0.638 Mpa·mm CI [−0.889, −0.406] Mpa·mm).

Cryopreservation did not affect the tests, neither histologically nor ultrastructurally (Figure 1D). Biomechanically, no significant differences were observed in either tensile or compression tests (Appendix A).

### 3.2. In Vivo Implantation

After implantation (Figure 2A), the rabbits were euthanized at the predefined time points. Graft loss was observed in 12.5% of cases due to self-extraction by the animal, two non-cryopreserved grafts (one at 2 weeks and one at 8 weeks), and two cryopreserved grafts (both at 8 weeks). All wound reopenings were detected within 6 h and were promptly cleaned, closed, and followed by prolonged antibiotic therapy. After sacrifice, macroscopic assessment showed that the surviving tracheas were completely integrated with the surrounding tissue, with abundant vascularization (Figure 2B).

#### 3.2.1. Histology

Histological analysis (Table 1, Figure 2C,D) showed no differences between cryopreserved and non-cryopreserved rabbit tracheas. There was an initial mild peak of inflammatory cellularity (eosinophils, neutrophils, plasma cells, and giant cells) with no differences between protocols. However, macrophage infiltration increased over time, especially in the cryopreserved tracheas. This increase was statistically significant (OR 10.487, CI [1.603–97.327]), but not clinically relevant, as it corresponded to a transition from “no macrophages” to “scant macrophages per 400× field” according to ISO grading. Macrophage infiltrates were observed mainly on the external tracheal face and in aggregates forming sheets, including fibroblasts in continuity with the cartilage, as a neo-perichondrium (Figure 2D).

Lymphocyte counts significantly decreased on a weekly basis after implantation (OR 0.049, [CI 0, 0.995]), thus indicating the absence of tissue rejection. This trend was unaffected by cryopreservation (Figure 2E).

Connective tissue formation increased progressively over time (OR 5.635; 95% CI: 1.941–20.296), regardless of cryopreservation. The occasional presence of adipose tissue or cartilage necrosis was not associated with any specific protocol. A trend toward increased capillary density was observed over time (mean capillary counts at weeks 2, 4, 8, and 12: 13.2, 20.4, 22.6, and 20.25, respectively; Figure 2F), but high inter-sample variability prevented statistical significance (Appendix A).

#### 3.2.2. Biomechanics

The initial biomechanical alterations observed after decellularization were progressively reversed following implantation, with grafts regaining properties comparable to native trachea (Figure 3). In tensile testing (Table 2), ε_max_ normalized by week 4, while σ_max_ and E returned to baseline by week 8 and remained stable thereafter. A transient decrease in W/Vol was noted only at week 4, attributable to an outlier with disproportionately high energy absorption (Appendix A). Cryopreservation had no impact on these findings.

Although the decellularization processes had a negative impact on compression test variables prior to implantation, these differences disappeared after only 2 weeks of implantation (Table 3 and Appendix A).

## 4. Discussion

Tracheal replacement is more complex than one might expect from a mere conduit, mainly because this organ must retain several important characteristics in order to maintain its function. An ideal substitute must be biocompatible, non-harmful, vascularized, and non-carcinogenic (due to the organ itself or to the need for immunosuppressants) [3,5]. In addition, the graft should be readily available, either as an off-the-shelf product or through a minimally delayed preparation process [4].

The aim of decellularization is to eliminate all cellular components capable of triggering alloimmune rejection while preserving the extracellular matrix, thus retaining all the inherent characteristics of the organ [14]. Various decellularization protocols have been described, all of which use cytotoxic elements (detergents, CO_2_, hyper- and hypotonic solutions) or some combination of these [13]. However, all these elements affect the matrix, at least to some extent [13].

Using the protocol described here, we successfully obtained decellularized tracheas that retained their histological and ultrastructural integrity. Biomechanical testing showed statistically significant decreases in σ_max_, E, and W/Vol, with no alterations in ε_max_. Radially, W/S was not altered, but a decrease was observed in f and ℛ. All these variations were statistically significant but clinically irrelevant and fully reversible following implantation.

Regarding tissue rejection, when it occurs in an implant, the cellular response is highly lymphocytic, accompanied by destruction of the implanted tissue [15]. In unrejected tracheal transplants, abundant infiltration by macrophages has been previously observed in the bibliography after 4 weeks [16], sometimes accompanied by the presence of monocytes and giant cells [17]. In general, habitual findings are an initial inflammatory reaction after implantation, which decreases over time [18].

Our findings are consistent with those previous reports involving implants that have been satisfactorily tolerated [16,17,18]. We detected a mild inflammatory peak at 2–4 weeks post-implantation, followed by a progressive decline, especially in lymphocytes, thus ruling out tissue rejection. The only increase observed was in the number of macrophages.

Initially, macrophages act in chronic inflammation and degradation of implant-derived materials. In such a scenario, multinucleated giant cells arise from proinflammatory (M1) macrophages. Subsequently, as demonstrated in bone implants, and under the influence of IL-4, IL-13, and α-tocopherol [19], the M1 phenotype diminishes and polarization toward the M2 phenotype occurs. This latter phenotype expresses mannose receptors (together with CD163 and CD209) and is associated with wound healing, angiogenesis, and tissue remodeling [20,21,22]. Taken together, these observations are consistent with the emergence of a reconstructive M2 phenotype in our samples. This hypothesis is supported by the decrease in other inflammatory cells, indicating a non-inflammatory microenvironment in which the infiltrate contained fibroblasts that formed laminated structures as an additional sign of the regenerative character of these tissues. Additionally, the biomechanical characteristics of the organ, rather than worsening—as would be expected in an inflammatory environment—appeared to reach and maintain their maximum values as the number of macrophages increased.

Loss of collagen and elastin is a known consequence of decellularization and cryopreservation, contributing to structural and biomechanical weakening of the graft [13], with a reported reduction of up to 62% in tracheal compression resistance [23], an effect also associated with cryogenic freezing of the organ [18]. However, implantation often induces progressive remodeling that restores mechanical function over time, although they may retain some alterations, especially radially [18]. In our model, the biomechanical characteristics behave comparably to native trachea within two (radially) to eight weeks (axially) after implantation.

Therefore, given that after 8 weeks, inflammatory infiltration is minimal and the biomechanical properties are optimal, we propose that an 8-week prelamination period is the most appropriate to achieve a fully vascularized, biomechanically functional graft suitable for definitive implantation. At this time point, the result is an organ functionally equivalent to the native trachea—encompassing, unlike other protocols, the entire circumference and exhibiting both histological and biomechanical equivalence—with the added advantage of being readily available, as cryopreservation is well tolerated.

This work represents a first step toward the development of a human tracheal substitute. Our model still has some limitations, such as the relatively low n resulting from adherence to the 3Rs (replacement, reduction, and refinement) mandated by the European Directive, the inherent losses associated with rabbit experimentation, and the fact that this was an ectopic implant. Therefore, although the tracheas demonstrated characteristics analogous to the native ones, in subsequent experiments we will evaluate their tolerance to exposure in the rabbit airway. The ultimate goal will be to reproduce this same tracheal replacement protocol in humans, using donor human tracheas.

## 5. Conclusions

The decellularization and cryopreservation protocol described here is effective, producing a tracheal replacement that retains the inherent characteristics of the native organ, with only minimal loss of certain properties. Moreover, the organ can be easily preserved for off-the-shelf use. These replacement constructs are fully biocompatible, well-integrated, and exhibit excellent tolerance upon implantation. By 8 weeks post-implantation, the graft recovers all baseline structural and functional characteristics, suggesting that this is the optimal duration for prelamination prior to eutopic tracheal implantation. Having identified the optimal prelamination period, the next step in the development of a suitable tracheal substitute is the eutopic implantation of a pedicled trachea.

## Figures and Tables

**Figure 1 biomedicines-13-02401-f001:**
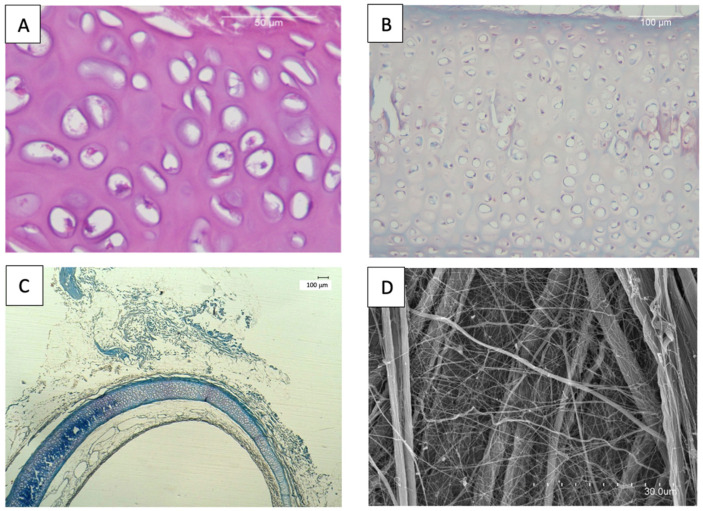
(**A**) H-E: Decellularized trachea with minimal debris. (**B**) O: Decellularized, cryopreserved cartilage analogous to native organ. (**C**) MT: Decellularized, cryopreserved tracheal cartilage. (**D**) SEM: Decellularized, cryopreserved trachea. Collagen fibers’ typical arrangement.

**Figure 2 biomedicines-13-02401-f002:**
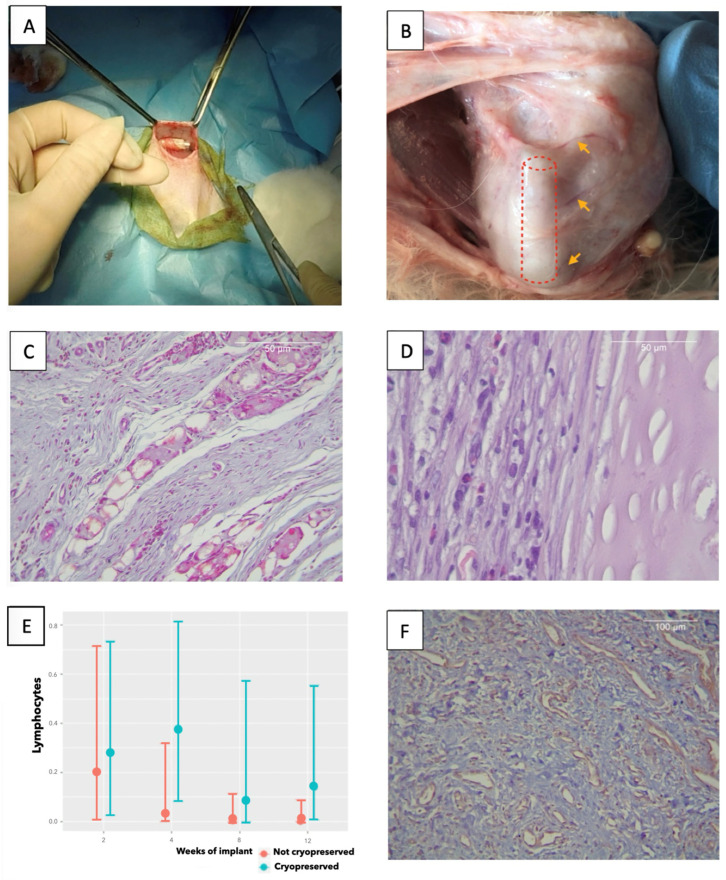
(**A**) Implantation surgery. (**B**) 8 week implant; vascularization. (**C**) MT: 4 week implant; giant cell sheets. (**D**) H-E: 12 week implant; macrophages and fibroblasts palisades. (**E**) Marginal differences; lymphocytes over time. (**F**) CD31; abundant vascularization.

**Figure 3 biomedicines-13-02401-f003:**
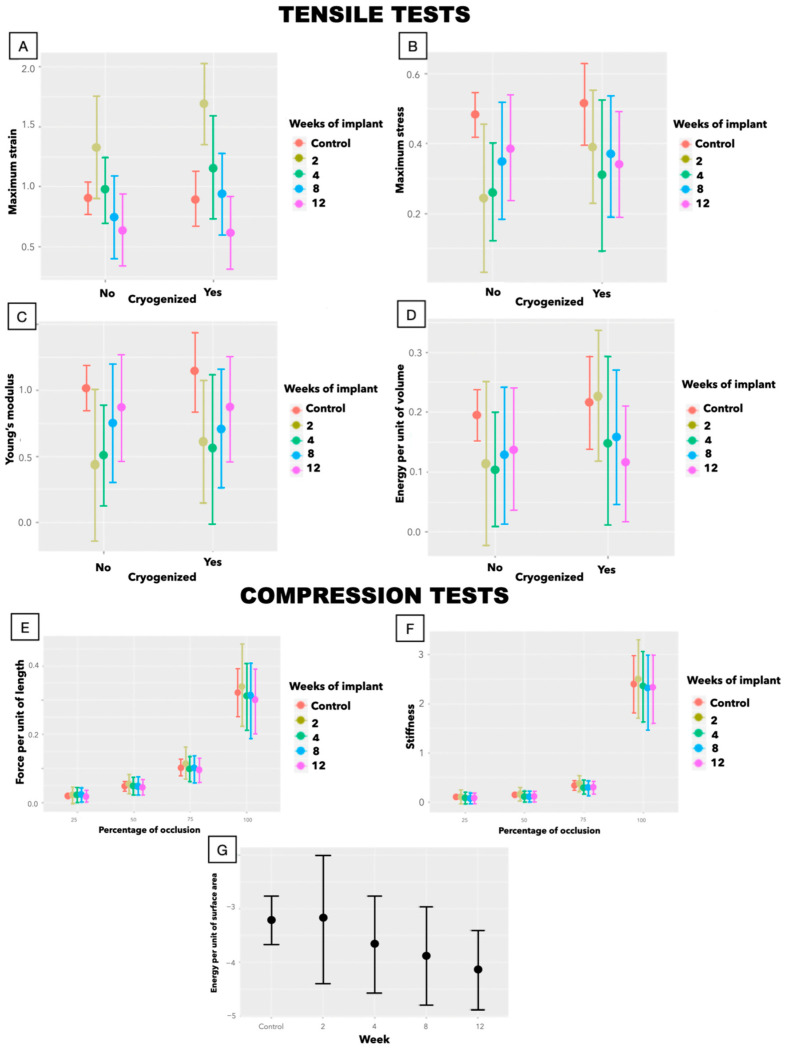
Marginal effects graphs: changes over time in (**A**) ε_max_, (**B**) σ_max_, (**C**) E, and (**D**) W/Vol depending on cryopreservation. Changes over time of (**E**) *ƒ*, (**F**) ℛ, and (**G**) W/S with the percentage of compression.

**Table 1 biomedicines-13-02401-t001:** Ordinal regression model of histologic variables. (Significant associations are shown in bold. Histologic values are expressed relative to non-cryopreserved tracheas, comparing cryopreserved tracheas, implantation time in non-cryopreserved tracheas, and implantation time in cryopreserved tracheas.)

		exp.Estimate	95% CI
EOSINOPHILS	Cryopreservation	1.818	0.089–31.706
	IT	0.51	0.201–1.25
	IT + Cryopreservation	0.772	0.22–2.973
NEUTROPHILS	Cryopreservation	1.593	0.049–37.667
	IT	0.333	0.069–1.242
	IT + Cryopreservation	1.314	0.18–9.402
LYMPHOCYTES	Cryopreservation	1.641	0.052–73.541
	**IT**	**0.049**	**0–0.995**
	IT + Cryopreservation	14.638	0.569–2517.091
PLASMOCYTE	Cryopreservation	1.34	0.053–25.632
	IT	0.358	0.068–1.354
	IT + Cryopreservation	1.437	0.229–9.54
MACROPHAGES	Cryopreservation	0.018	0–1.174
	**IT**	**0.29**	**0.079–0.891**
	**IT + Cryopreservation**	**10.487**	**1.603–97.327**
GIANT CELLS	Cryopreservation	0.261	0.018–3.957
	IT	1.152	0.432–2.906
	IT + Cryopreservation	2.09	0.528–8.518
CONNECTIVE TISSUE	Cryopreservation	3.833	0.322–52.99
	**IT**	**5.635**	**1.941–20.296**
	IT + Cryopreservation	1.175	0.325–4.834
FATTY TISSUE	Cryopreservation	0.003	0–2.747
	IT	0.214	0.001–2.392
	IT + Cryopreservation	17.633	0.606–144618.768
NECROSIS	Cryopreservation	2.079	0.154–27.353
	IT	0.97	0.369–2.7
	IT + Cryopreservation	1.105	0.299–3.943
VASCULARIZATION	Cryopreservation	1.146	0.696–1.841
	IT	1.14	0.913–1.356
	IT + Cryopreservation	0.919	0.741–1.172

(CI: credible interval; IT: implantation time).

**Table 2 biomedicines-13-02401-t002:** Results of traction tests in the implanted tracheas. (Significant associations are shown in bold.)

		Estimate	95% CI
$\varepsilon_{max}$	Intercept	0.902	0.773–1.032
	**2 weeks**	**0.426**	**0.012–0.847**
	4 weeks	0.074	−0.17–0.323
	8 weeks	−0.156	−0.479–0.16
	12 weeks	−0.268	−0.541–0.004
	2 w + cryopreservation	0.371	−0.133–0.868
	4 w + cryopreservation	0.186	−0.293–0.675
	8 w + cryopreservation	0.202	−0.245–0.64
	12 w + cryopreservation	−0.012	−0.393–0.347
σ_max_	Intercept	0.481	0.418–0.544
	**2 weeks**	**−0.236**	**−0.438–−0.037**
	**4 weeks**	**−0.221**	**−0.349–−0.096**
	8 weeks	−0.135	−0.287–0.02
	12 weeks	−0.096	−0.233–0.042
	2 w + cryopreservation	0.112	−0.141–0.359
	4 w + cryopreservation	0.016	−0.222–0.252
	8 w + cryopreservation	−0.01	−0.214–0.209
	12 w + cryopreservation	−0.076	−0.256–0.105
E	Intercept	1.017	0.845–1.187
	**2 weeks**	**−0.587**	**−1.144–−0.035**
	**4 weeks**	**−0.512**	**−0.855–−0.184**
	8 weeks	−0.267	−0.676–0.148
	12 weeks	−0.145	−0.516–0.215
	2 w + cryopreservation	0.057	−0.602–0.735
	4 w + cryopreservation	−0.076	−0.699–0.523
	8 w + cryopreservation	−0.17	−0.727–0.388
	12 w + cryopreservation	−0.134	−0.614–0.358
WVol	Intercept	0.195	0.152–0.238
	2 weeks	−0.081	−0.212–0.054
	**4 weeks**	**−0.09**	**−0.173–−0.006**
	8 weeks	−0.067	−0.173–0.038
	12 weeks	−0.059	−0.147–0.034
	2 w + cryopreservation	0.092	−0.075–0.256
	4 w + cryopreservation	0.023	−0.13–0.177
	8 w + cryopreservation	0.009	−0.127–0.15
	12 w + cryopreservation	−0.042	−0.165–0.077

CI: credible interval; W/Vol: energy stored per unit of trachea volume (mJ·mm^−3^); ε_max_: maximal strain (unitless); σ_max_: maximal stress (MPa); w: weeks; E: Young’s modulus (MPa).

**Table 3 biomedicines-13-02401-t003:** Results of compression tests in implanted tracheas.

	Variables	Estimate	Lower.95.
f	Intercept	0.019	0.01–0.029
	2 weeks	0.006	−0.026–0.04
	4 weeks	−0.004	−0.03–0.02
	8 weeks	−0.005	−0.039–0.016
	12 weeks	−0.006	−0.033–0.011
	Cryopreservation	−0.005	−0.016–0.006
R	Intercept	0.101	0.053–0.149
	2 weeks	0.027	−0.17–0.228
	4 weeks	−0.009	−0.207–0.141
	8 weeks	−0.029	−0.257–0.101
	12 weeks	−0.023	−0.187–0.078
	Cryopreservation	0.027	−0.17–0.228
WVol	Intercept	−2.683	−3.14–−2.23
	2 weeks	0.052	−1.201–1.343
	4 weeks	−0.436	−1.503–0.665
	8 weeks	−0.661	−1.718–0.425
	12 weeks	−0.902	−1.822–0.051
	Cryopreservation	−0.28	−0.85–0.287

## Data Availability

The original contributions presented in this study are included in the article/Appendix A, further inquiries can be directed to the corresponding authors.

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
