# Peer review of "Experimental Airway Allogenic Transplantation Model with Decellularized Cryopreserved Tracheas"

_biomedicines, 2025, doi:10.3390/biomedicines13102401_

Round 1
Reviewer 1 Report
Comments and Suggestions for Authors
The manuscripts describes some newer techniques to increase the cellularization of a tracheal implant that is devoid of any cells, and the techniques presented appear to demonstrate that their way of handling and treatment can result in a tracheal segment that appears to closely mimic the intact airway. The paper is well written and methods appropriately described, and the rigor of the studies is excellent. Comments and concerns are generally minor.
- The authors, in the introduction section, should explicitly describe how their methods differ from what has been done previously, and in the discussion section explicitly explain the improvements over the previous work in the area.
- The big increase in macrophages vs intact and cryopreserved tracheas is dealt with in a cursory manner. The authors should enhance their discussion of this anomoly (vs other inflammatory/immune cells in the discussion at the top of page 15. Specifically, they should address whether this could be some sort of immune response, whether the macrophages might be responding to what they perceive as a foreign substance, etc. as this could be a key finding.
- It's unclear how the authors are postulating as to exactly how this procedure can be translated to humans with disease who are receiving a tracheal graft. Are they suggesting that the decellularized rabbit trachea can be used in humans, or that autopsy or biopsy tissue from other humans can be utilized in this manner? Whatever the goal here, the manuscript requires this kind of discussion, as it is entirely unclear from the present manuscript where they wish to take this.
2.
Author Response
We would like to sincerely thank you for the careful evaluation of our manuscript and for the constructive comments and suggestions provided. We greatly appreciate the time and expertise invested in your review, which have been very helpful in improving the clarity, accuracy, and overall quality of our work. Please find attached the PDF file, in which we address each of your comments point by point and describe the corresponding changes implemented in the manuscript, along with the updated version of the paper.

Reviewer 2 Report
Comments and Suggestions for Authors
Dear Author,
First of all thank you for submitting your research results to the Journal of Biomedicines.
The manuscript you submitted addresses the important and still unresolved challenge of tracheal replacement in airway pathology. The authors present a decellularization and cryopreservation protocol that is subsequently tested in vivo using a rabbit model. In a prospective experimental design, tracheal scaffolds are prepared, sterilized, and supported with stents, then implanted with bilateral fasciomuscular flaps. The study evaluates histological and biomechanical outcomes over different time points, comparing cryopreserved with non-cryopreserved constructs. The findings suggest that decellularization is effective without compromising biomechanical integrity, and that the implanted scaffolds are well tolerated, vascularized, and gradually recover native-like histological and biomechanical features within 8 weeks, regardless of cryopreservation status. Overall, the study proposes a potentially relevant strategy for generating biocompatible tracheal grafts.
Please find my review attached as PDF format.
Sincerely,
Reviewer.

Author Response
We are very grateful for your thorough review of our manuscript and for the insightful remarks you have shared. Your comments, including the suggestion to improve the quality of the figures, have been extremely valuable in helping us refine both the presentation and the content of the work. We have carefully addressed each of your observations and detailed in the attached file the modifications made in response to your feedback, together with the updated version of the manuscript.

Round 2
Reviewer 2 Report
Comments and Suggestions for Authors
Dear Authors,
I appreciate your answer and changes of the manuscript!
The paper now has a form and quality to be published.
Best wishes,
Reviewer 2